# Continuous Growth Monitoring and Prediction with 1D Convolutional Neural Network Using Generated Data with Vision Transformer

**DOI:** 10.3390/plants13213110

**Published:** 2024-11-04

**Authors:** Woo-Joo Choi, Se-Hun Jang, Taewon Moon, Kyeong-Su Seo, Da-Seul Choi, Myung-Min Oh

**Affiliations:** 1Division of Animal, Horticultural and Food Sciences, Chungbuk National University, Cheongju 28644, Republic of Korea; ujuchoe79@chungbuk.ac.kr (W.-J.C.); zsh8976@naver.com (S.-H.J.); nari4491@naver.com (K.-S.S.); daseul7312@gmail.com (D.-S.C.); 2Smart Farm Research Center, Korea Institute of Science and Technology (KIST), Gangneung 25451, Republic of Korea; tmoon.hort@kist.re.kr

**Keywords:** artificial intelligence, computer vision, crop growth, indoor farming, lettuce

## Abstract

Crop growth information is collected through destructive investigation, which inevitably causes discontinuity of the target. Real-time monitoring and estimation of the same target crops can lead to dynamic feedback control, considering immediate crop growth. Images are high-dimensional data containing crop growth and developmental stages and image collection is non-destructive. We propose a non-destructive growth prediction method that uses low-cost RGB images and computer vision. In this study, two methodologies were selected and verified: an image-to-growth model with crop images and a growth simulation model with estimated crop growth. The best models for each case were the vision transformer (ViT) and one-dimensional convolutional neural network (1D ConvNet). For shoot fresh weight, shoot dry weight, and leaf area of lettuce, ViT showed *R*^2^ values of 0.89, 0.93, and 0.78, respectively, whereas 1D ConvNet showed 0.96, 0.94, and 0.95, respectively. These accuracies indicated that RGB images and deep neural networks can non-destructively interpret the interaction between crops and the environment. Ultimately, growers can enhance resource use efficiency by adapting real-time monitoring and prediction to feedback environmental controls to yield high-quality crops.

## 1. Introduction

Crop growth is the result of cumulative responses to growth environments such as light, temperature, and water. An adequate cultivation environment guarantees high crop productivity and quality [1,2]. Previous studies have reported on crop responses to single and multiple environmental factors [3,4]. However, cumulative changes following the crop developmental stage are still an uncharted territory because of their intricacy.

Dynamic feedback control based on a crop’s developmental stage can enhance the resource use efficiency of crop production systems. In particular, the productivity of indoor farming is highly dependent on the efficiency of environmental controls [5]; in this case, the dynamic control can significantly reduce input costs [6]. Dynamic data collection should precede feedback control to obtain appropriate feedback. However, conventional destructive investigations cannot track the same target crops, and the data interval cannot be shortened significantly. Some studies have attempted to extract optimal crop environments using non-destructive factors, such as photosynthesis and fluorescence [7,8,9]; however, these targets are not directly related to the crop product and quality. Thus, crop properties that are directly related to productivity, such as leaf area, fresh weight, and dry weight, should be monitored and collected for dynamic feedback control.

Recently, computer vision technology based on deep learning, including agricultural applications, has been developed [10,11]. Image sensors, such as RGB, hyperspectral, and thermal imaging, can efficiently collect continuous data non-destructively without spatial restrictions [12,13,14]. Horticultural applications using convolutional neural networks and crop images have been employed for tasks such as pest and disease diagnosis, fruit detection, growth quantification, and growth estimation [15,16,17,18]. Thus, convolutional neural networks can be used to monitor target crops and extract related growth factors from images for dynamic feedback control.

The aim of this study was to propose a non-destructive crop-growth prediction method in response to environmental changes based on the consecutive acquisition of lettuce images from indoor farming for growth parameter extraction, shoot fresh weight (SFW), shoot dry weight (SDW), and leaf area (LA), which are highly related to the productivity and quality of leafy vegetables. The limited data were divided into two datasets for growth parameter extraction from the images and growth prediction using the extracted growth factors. Various deep neural networks for computer vision were compared regarding their growth-extraction performance using continuously captured lettuce images. The extracted growth parameters were estimated using various similar networks for sequence data interpretation, and their performances were compared.

## 2. Materials and Methods

### 2.1. Workflow

The ultimate objective was model construction to predict crop growth for the entire cultivation period using only environmental data; however, securing sufficient growth data based on destructive investigation would incur too much cost. Therefore, target crop-growth factors were estimated using an image converter (Figure 1). A trained and verified model converting images into growth factors generated sufficient data to train the growth predictor. Finally, the trained growth predictor was validated.

### 2.2. Data Collection

Data loggers (ETD-H004, N.THING Inc., Seoul, Republic of Korea) installed at the crop level collected the temperature, relative humidity, and CO_2_ concentration at one-minute intervals. The collected data were preprocessed to train the deep-learning model. Crop images were collected in 1st–3rd cultivations at consistent intervals. The same crops were destructively investigated after the images were acquired. Imaging data were collected using a dark box with a white LED (560 × 560 × 560 mm, L × H × W). The crops were placed in the middle of a dark box, and top-view images (3024 × 3024 pixels) were captured using a Galaxy S20 (Samsung Electronics Inc., Suwon, Republic of Korea). The distance between the camera and the bottom of the lettuce was set to 50 cm. SFW was then measured using a precision balance (SI-234; Denver Instruments, Denver, CO, USA) and LA was measured using the LI-3100C leaf area meter (Li-COR, Lincoln, NE, USA). The destructed shoots were dried for 72 h at 70 °C using forced-air drying oven (VS-1202D3, Vision, Daejeon, Republic of Korea) to determine dry weight data. A total of 161 images and growth datasets were collected.

In the 4th–9th cultivations, continuous top-view images of the target lettuce were collected using a Raspberry Pi RPI 8MP Camera and Raspberry Pi 4 Model B 4GB (Raspberry Pi Foundation, Caldecote, South Cambridgeshire, UK) at 20-min intervals. The collected images were 500 × 500 pixels, and the red color cast due to the warm-white LEDs was removed using an auto-white balance from the Picamera library. Certain non-target crops were destructively investigated to measure the growth parameters for validation during the 4th and 5th cultivation periods. The number of continuous images was 2016 for the six crops.

### 2.3. Data Preprocessing

All images were resized to 500 × 500 pixels to ensure consistency with the continuous images from the Raspberry Pi. The discontinuous data from the dark box before the destructive investigation were divided into training (85%) and testing (15%) datasets. The training datasets were augmented with a synthetic background using Rembg and OpenCV libraries. The augmented training dataset increased 9-fold, and the number of augmented data was 1224. Minimum-max normalization was conducted for the collected images and growth data.

The environmental data for the continuous image datasets were resampled and averaged in a 20-min interval and then matched to the estimated growth data from the continuous images. The datasets were grouped into 2-day lengths to predict daily growth; the input was environment and growth pairs with days after transplanting values, and the output was daily growth. The datasets were resampled at intervals of 20 min, 1 h, 2 h, and 3 h to determine the optimal interval for growth prediction. The last value of the day was set as the daily growth (Figure 2). A total of 285 datasets were used. The datasets were sampled randomly and divided into training (75%), validation (15%), and verification (10%) groups.

### 2.4. Image Converter Models for Growth Estimation Using Images

Image converters were trained to translate the crop images into growth parameters: SFW, SDW, and LA. A multitask learning (MTL) structure was adopted to estimate these three growth parameters. Common layers based on the MTL can assist in the generalization ability of the trained models [19]. Feed-forward neural network (FFNN), a traditional deep learning algorithm, convolutional neural network (ConvNet), widely used for image analysis, and vision transformer (ViT) algorithms [20,21] used for the state-of-the-art image analysis algorithm were applied and compared after five training sessions with 300 epochs (Table 1).

The input was flattened for the FFNN. The feature extraction of ConvNet was conducted with pre-trained layers from the Keras library: VGG19 [22], ResNet [23], MobileNet [24], Inception-v3 [25], Xception [26], EfficientNet [27], ViT for small datasets (ViT-s) [28], Swin Transformer [29], Conv-Mixer [30], and vanilla ViT were trained and compared as ViT models. The hyperparameters were empirically adjusted for all the models.

The loss function was the mean absolute error (MAE) for the regression task, the optimizer was adamW, and the dropout rate was 0.3. The activation function for the FFNN and ConvNet was a rectified linear unit (ReLU), and that for ViT was a Gaussian error linear unit (GELU). The number of layers and perceptrons in the MTL layer have been determined using various combinations to efficiently utilize resources and achieve high performance. The final output layer was linear. The performance of the trained models was evaluated using *R*^2^, the root mean square error (RMSE), and the normalized RMSE (NRMSE).

### 2.5. Growth Predictor Models for Growth Prediction Using Extracted Growth Factors

The target task of the growth predictors was to estimate crop growth on any day t during the cultivation period. The target outputs SFW, SDW, and LA were estimated using the best image converter from the previous model training. To predict the growth of the next day, the data inputted into the model includes three estimated growth parameters from the previous model, three environmental data types, crop age, and information at 20-min intervals over two days. Similarly, the MTL structure was designed and used to predict the three growth parameters. The FFNN, long short-term memory (LSTM) [31], bidirectional LSTM (BiLSTM) [32], and one-dimensional ConvNet (1D ConvNet), used for time-series prediction, were selected as growth predictors (Table 2). The vanilla 1D ConvNet was compared to the 1D ConvNet with U-Net feature extraction [33]. The models were trained five times, and their average performances were compared.

### 2.6. Computation

The data and model were computed on a server with an i9-12900k processor (Intel, Santa Clara, CA, USA) and RTX A6000 D6 48GB (NVIDIA, Santa Clara, CA, USA). The model was trained with Keras in Python 3.8 on Ubuntu 20.04.

### 2.7. Plant Materials and Growth Condition

Lettuce (*Lactuca sativa* L. cv. *Caipira*) was selected and grown in an indoor farming system under artificial light for nine cultivation periods. Seeds (Enza Zaden, Enkhuizen, The Netherlands) were sown in 60-hole trays containing sponge cubes (35 × 35 × 30 mm, L × W × H). Photosynthetic photon flux density (PPFD) of 200 μmol m^−2^ s^−1^ with warm-white LEDs and 24 h of daytime, temperature of 24 °C, and relative humidity of 90% were set for the first few days for germination. Distilled water was sub-irrigated at that time. The number of crops in a layer was 28. Sonneveld nutrient solution (Sonneveld and Straver, n.d.) and Hoagland nutrient solution [34] and Hoagland nutrient solution alone [35] were sub-irrigated for 1st–3rd and 4th–9th cultivations, respectively, after rooting was observed under the cubes. The nutrient concentration was 0.8 dS m^−1^ and the pH was 5.8. The seedlings were then transplanted into indoor farming modules with a semi-NFT water channel (3540 × 80 × 60 mm, L × W × H) 11 days after germination. The transplanted lettuces were cultivated for 4 weeks with PPFD of 240 μmol m^−2^ s^−1^ and 16 h of light; temperature and humidity were maintained at 20–23 °C and 80–90%, respectively; and CO_2_ concentration was 500–700 μmol mol^−1^.

## 3. Results

### 3.1. Image Converter Using Computer Vision Technology and Top-View Images

The best structures for ConvNet and ViT were EfficientNet and ViT, respectively (Appendix A). ViT-s exhibited the best performance, with the highest R^2^ and the lowest RMSE and NRMSE among the selected models (Figure 3). The input channels, image size, and patch size were also optimized for ViT-s to yield reliable results for training the growth estimators. The best combination was an image size of 100 × 100 and a patch size of 10 × 10 using all the RGB-mask channels. The trained ViT-s with the optimized parameters showed *R*^2^ values of 0.89, 0.78, and 0.93 for SFW, SDW, and LA, respectively (Figure 4).

All models tended to estimate LA accurately but dry weight was not accurately estimated. The models showed a high *R*^2^ for LA; however, the RMSE and NRMSE were only relatively high. FFNNs are often used as an acceptable model, although they are basic and simple algorithms for deep learning. However, the trained FFNN failed to estimate all target outputs. Overall, the estimated growth parameters with the ViT-s were acceptable; therefore, the trained ViT-s were used to extract growth parameters from the images to train growth predictors.

### 3.2. Growth Predictor with Sequence Interpretation Algorithms

ViT-s estimated SFW, SDW, and LA from lettuce images of 4th–9th cultivations to create continuous growth datasets for growth predictors (Figure 5). Growth tendencies vary under different environmental conditions. This tendency followed a sigmoidal function, which is known as the standard growth curve [36]; the extracted growth parameters were reasonably distributed, although these values could not be compared to the real measured values.

The extracted growth parameters exhibited a natural growth tendency without severe fluctuations. The growth predictors were trained to predict the next growth parameters from the previous environment and growth data. The best structures for the LSTM and 1D ConvNet were the vanilla LSTM and 1D ConvNet with the U-Net feature extractor (Appendix A). Consequently, the 1D ConvNet with U-Net exhibited the best performance (Figure 6). LSTMs could not produce acceptable results, despite being frequently used for sequence interpretation.

Similar to the image converter, the input property for the 1D ConvNet with U-Net was optimized, and the 2-h interval yielded the best performance. The trained 1D ConvNet with U-Net based on the optimized parameters showed *R*^2^ values of 0.96, 0.95, and 0.94 for SFW, SDW, and LA, respectively (Figure 7). The 1D ConvNet with U-Net could predict all growth factors with high accuracy. In contrast to the image converter, the trained FFNN yielded accurate growth factors, although its accuracy was lower than that of the trained 1D ConvNet. In particular, the image converter showed relatively low accuracy for LA, but the growth predictor showed accurate predictions for the estimated LA.

## 4. Discussion

### 4.1. Result Analysis Based on Data Condition

In this study, the image converter and growth predictors were trained separately using a non-destructive data collection process. The image conversion is also an estimation based on the regression; therefore, some empirical techniques, such as hyperparameter setting and avoiding overfitting, are required to generalize the suggested training steps [37]. We attempted to create general datasets by collecting data from several independent cultivations to ensure the versatility of the model. The gaps in the different cultivations seemed to overcome existing data augmentation processes, such as background replacement. Therefore, foreseeing considerations are required during the data preparation step to build a robust model.

The image converter estimated SDW at a lower accuracy than that of SFW and LA. This appears to be due to the different distributions of the training and verification datasets because of the different nutrient solutions. Nutrient solutions can change the mineral and water contents of crops, which are difficult to sense with RGB images [38], resulting in an underestimation of SDW (Figure 3). An accurate estimation of hidden growth factors, such as SDW, should use different sensors, such as near-infrared cameras [39]. Mathematical crop modeling or simple regression methods based on environmental data can also be adopted [40,41]. Similarly, according to the high *R*^2^ and RMSE values, the image converter accurately estimated the LA growth tendency; however, it was difficult to measure accurate values. Securing large datasets is a definite solution; however, data are limited in most agricultural cases. Practical solutions include additional image processing, such as adding contours based on domain knowledge. In general, the trained ViT-s demonstrated adequate accuracy; therefore, the model was used as an image converter for growth prediction.

### 4.2. Perspectives of Deep Learning Modeling

Three core algorithms, FFNN, ConvNet, and ViT, were selected and compared with the image converters. The best average *R*^2^ value was 0.87, which was acceptable for extracting growth data for growth predictors. ConvNets have been frequently applied to horticultural image data in previous studies [42]; however, it seems that the global attention mechanism of ViT-s is more suitable for regression tasks using images than the local receptive fields of ConvNets [43]. The locality inductive bias resulting from small amounts of data can also be avoided with shifted patch tokenization and local self-attention in the ViT-s approach. Therefore, ViT algorithms can replace ConvNets for computer vision tasks, even for agricultural data.

The low accuracy of the LSTM for growth prediction can also be explained from a modeling perspective. LSTM is a powerful tool for interpreting sequence data. However, hourly and daily interval differences are difficult for LSTM to match using only simple sequence data. LSTM often performs better when it comes to time-series data based on its autoregressive structure. However, the hyperparameters should be sophisticatedly adjusted to interpret data with high disagreement in the input and output sequences.

Growth prediction appeared to be an easy task owing to the high accuracy of the trained FFNN (Figure 6). Similarly, dividing a task into smaller tasks is better than the end-to-end model approach when creating a robust model that can be interpreted by humans, which is called explainable artificial intelligence. Considering the various errors in small datasets, the divided tasks can ensure the existence of baselines. This could help enhance model reliability.

The 1D ConvNet appears to have advantages for these simple tasks. LSTM species requiring fewer parameters, such as gated recurrent units, can also be used [44]. However, data distribution is difficult to judge for small data conditions, and deep-learning algorithms are relatively easy to apply to new data; therefore, it is desirable to compare diverse algorithms to develop an adequate prediction model.

The high accuracy of the three growth parameters demonstrates the reasonableness of the predicted growth. However, hidden growth factors, such as dry weight, have a high variation; therefore, the given input data may not reflect all the variations in the output. In this study, SDW can be regarded as having low certainty based on the accuracy of the image converter; however, end-to-end prediction can yield a high accuracy similar to that of the growth predictor. Therefore, trained models should be validated with diverse perspectives for small datasets.

### 4.3. Domain Knowledge-Based Interpretation

Because crop growth can vary depending on the physiological properties of individual crops, predicting crop growth should consider physiological and genomic responses in conjunction with the developmental stage and cultivation conditions [45,46]. In this study, it was assumed that today’s crop growth can be represented with yesterday’s environment and crop growth. However, crop growth could be totally different due to events from the distant past, although the crop growth at any time t could be highly related to that at time *t − 1*.

If sufficient data can be collected, crop growth can be completely predicted using simulator-learned whole cultivation at once. Utilizing existing crop models can also enable deep-learning models to learn domain knowledge. Growth prediction for fruits and vegetables that can be affected by long-term environmental changes requires more complicated models [47]. However, the results showed that the basic growth parameters, SFW, SDW, and LA, could be predicted using continuous images and environmental data under limited conditions, such as leafy vegetables.

## 5. Conclusions

In this study, the SFW, SDW, and LA of lettuce under indoor farm conditions were predicted using crop data and top-view images with an image converter and growth predictor. The image converter was used to collect growth data from the RGB images, which non-destructively contain continuous SFW, SDW, and LA. The growth predictors were then trained to predict the next SFW, SDW, and LA based on previous growth and environmental data. The best models were ViT-s for the image converter and 1D ConvNet with U-Net as the growth predictor. These two-step processes require training the two models separately so that users can interpret and analyze each step for maintenance; however, the inference process after the training completion operates as an end-to-end model. Therefore, this process can be adapted to other crops and cultivation conditions through simple fine-tuning. Finally, the training protocol leads to real-time monitoring and growth quantification, thereby supporting dynamic feedback control. Image converters can be made more robust with hardware to sense hidden features such as dry mass and chlorophyll content in further studies.

## Figures and Tables

**Figure 1 plants-13-03110-f001:**
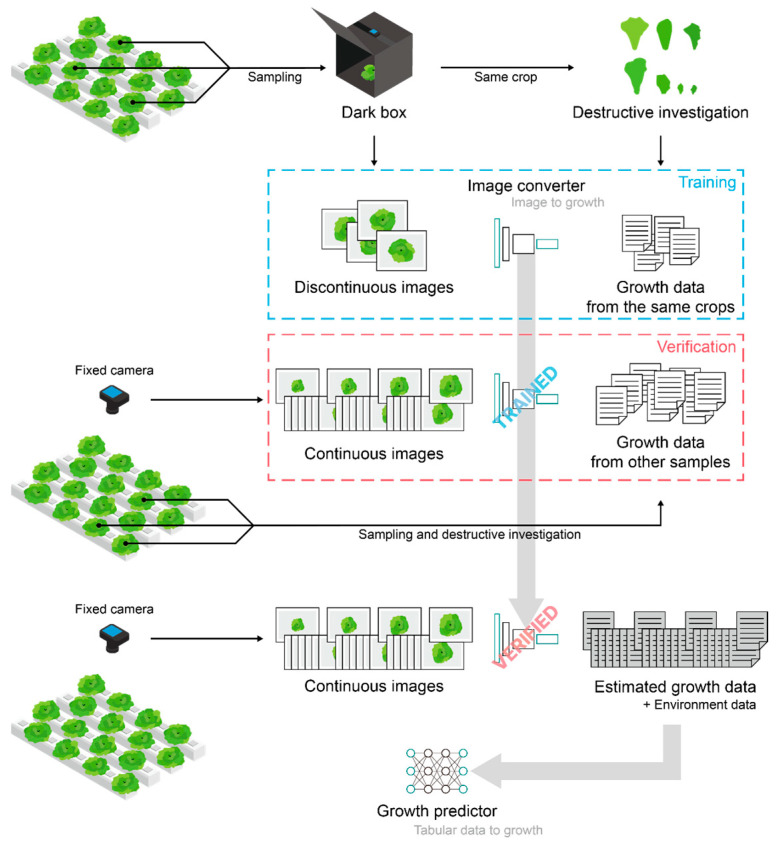
Workflow from data collection to image converter and growth predictor model construction.

**Figure 2 plants-13-03110-f002:**
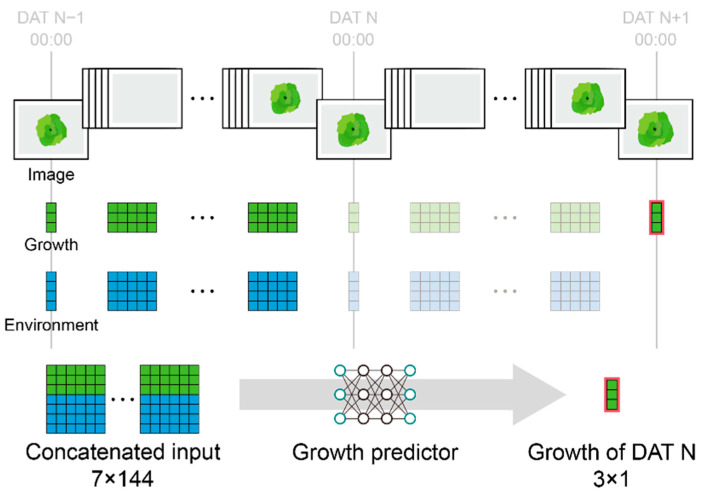
Input and output structure for growth predictors. The images were converted to growth factors using a well-trained image converter; 144 indicates information for two days, three times an hour at 20-min intervals.

**Figure 3 plants-13-03110-f003:**
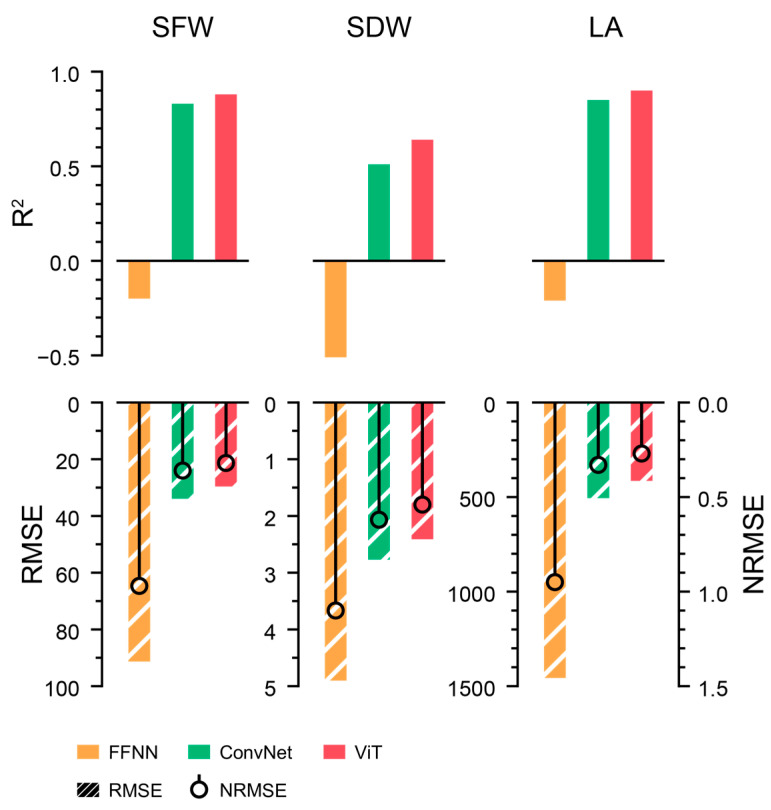
Accuracies of estimated growth parameters from the trained image converters. The units for RMSEs were omitted.

**Figure 4 plants-13-03110-f004:**
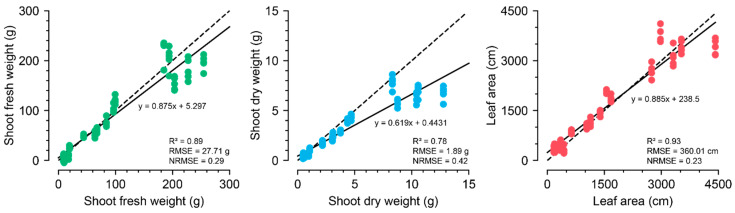
The growth-extraction results from images using ViT-s with the optimized input property. All five training results are represented in the figures. The specified *R*^2^, RMSE, and NRMSE represent the best performance among the five training runs. The dashed and solid lines represent *y = x* and the regression lines, respectively.

**Figure 5 plants-13-03110-f005:**
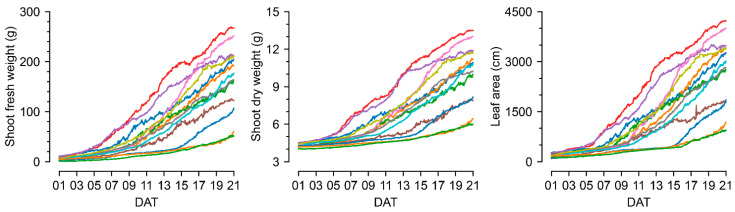
The estimated shoot fresh weight (SFW), shoot dry weight (SDW), and leaf area (LA) from top-view images using the trained image converter, ViT-s according to the days after transplanting (DAT). The same color represents the same cultivation.

**Figure 6 plants-13-03110-f006:**
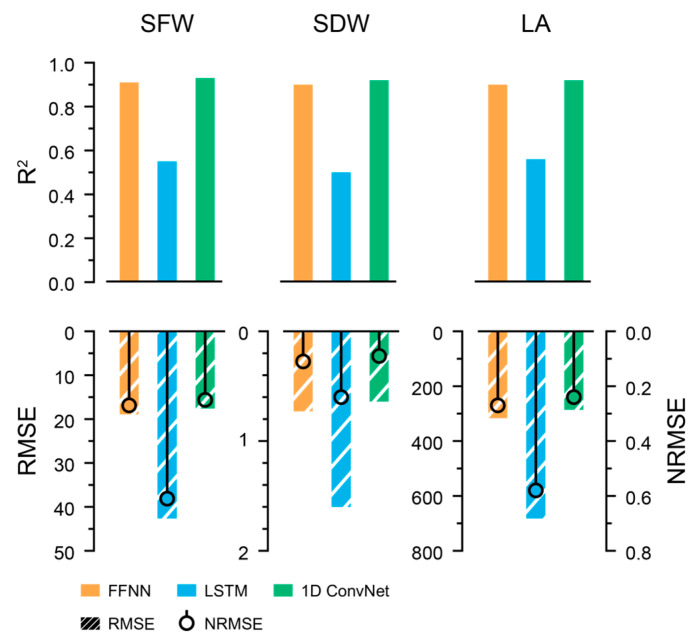
Accuracies of predicted growth factors from the trained growth predictors. The units for RMSEs were omitted.

**Figure 7 plants-13-03110-f007:**
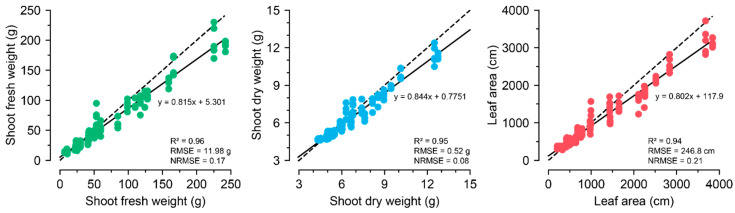
The growth prediction results from 1D ConvNet with U-Net feature extractor with the optimized input property. All five training results are represented in the figures. The specified *R*^2^, RMSE, and NRMSE represent the best performance among the five training runs. The dashed and solid lines represent *y = x* and the regression lines, respectively.

**Table 1 plants-13-03110-t001:** Structures of the feed forward neural network (FFNN), convolutional neural network (ConvNet), and Vision Transformer (ViT). Dense is a fully connected layer. Parameters were represented as “{type of layer}-{number of nodes in the layer}”. Feature extraction layer represents imported pretrained layers. MTL layer represents separated multitask layers, which were same for the three target growth factors.

Algorithms	FFNN	ConvNet	ViT
Input size	500 × 500 × 3		
Layer	Flatten Dense-2048 Dense-2048	Feature extraction layer Global average pooling	Cut patches (25 × 25) Linear projection-50 Positional embedding Multihead attention-8 Flatten
MTL layer	Dense-256 Dense-256	Dense-512 Dense-512 Dense-128 Dense-128	Dense-2048 Dense-1024
Outputs	Dense-1		

**Table 2 plants-13-03110-t002:** Structures of growth predictors: FFNN, long short-term memory (LSTM), bidirectional LSTM (BiLSTM), and one dimensional (1D) ConvNet. LSTM represent LSTM or BiLSTM cells. Refer to Table 1 for parameter representation.

Algorithms	FFNN	LSTM and BiSLTM	1D ConvNet
Input size	144 × 7		
Layer	Flatten Dense-1024 Dense-1024	LSTM-1024 Layer normalization	Feature extraction layer Flatten Dense-2048 Dense-2048
MTL layer	Dense-512 Dense-512	Dense-512 Dense-512	Dense-512 Dense-64
Outputs	Dense-1		

## Data Availability

The original contributions presented in this study are included in the article/Appendix A. Further inquiries can be directed to the corresponding author.

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
