# Peer review of "Continuous Growth Monitoring and Prediction with 1D Convolutional Neural Network Using Generated Data with Vision Transformer"

_plants, 2024, doi:10.3390/plants13213110_

Round 1
Reviewer 1 Report
Comments and Suggestions for Authors
This article proposes a method that uses Low-cost RGB images and computer vision to perform a non-destructive prediction crops growth. Various deep learning models are used to construct image-to-growth model and growth simulation model. The content of the background and the related works are well presented and the work is well motivated. There is well crafted experimental setup and a clear description of the data used. Addressing the following would improve the quality of the paper:
- The formulation of the solution is clear and the formulation of the solution somewhat understandable, the choice of the models used need more explanation. No detail or minimal details are provided for the models used. Particularly, how were the parameters were set.
- There must be a comparison to the state-of-the art
- The authors must also clear list the contributions made by the paper.
Comments on the Quality of English LanguageThere very minor errors that need to be corrected. In my opinion, the article reads well and flows.
Author Response
Comments 1: The formulation of the solution is clear and the formulation of the solution somewhat understandable, the choice of the models used need more explanation. No detail or minimal details are provided for the models used. Particularly, how were the parameters were set.
Respones 1: The models used for regressing growth parameters through image analysis were FFNN, CNN, and ViT. To identify the most suitable algorithm structure, six different CNN models and four ViT models were compared (lines 126-130). Detailed algorithm structures are described in lines 136-140. The models used for predicting growth were FFNN, LSTM, and 1D ConvNet, with two variations of LSTM and two of 1D ConvNet compared to find the optimal structure. Detailed algorithm structures for these models are provided in lines 142-153. The method for setting the number of layers and perceptrons in the MTL layer is additionally described in lines 134-136 and 150.
Comments 2 : There must be a comparison to the state-of-the art.
Respones 2 : In this study, the techniques explored range from traditional feedforward neural networks to CNN, widely used for image-based analysis, and the Vision Transformer, a state-of-the art method in image analysis. Additionally, Bi-LSTM and 1D ConvNet, both widely utilized for time-series analysis, were employed. The information provided, including your insights, has been clarified in lines 121-125 and 149-151.
Comments 3 : The authors must also clear list the contributions made by the paper.
Respones 3 : CRediT authorship contribution statement
Woo-Joo Choi: Conceptualization, Formal analysis, Investigation, Methodology, Software, Validation, Visualization, Writing – original draft, Writing – review & editing. Se-Hun Jang: Investigation. Kyeong-Su Seo: Investigation. Taewon Moon : Validation, Visualization, Writing – original draft, Writing-review & editing. Da-Seul Choi: Investigation. Myung-Min Oh: Conceptualization, Formal analysis, Data curation, Visualization, Funding acquisition, Project administration, Supervision, Writing – review & editing. All authors have read and agreed to the published version of the manuscript.
Reviewer 2 Report
Comments and Suggestions for Authors
The paper proposes a framework for non-destructive growth prediction of crops using images of the crops. This ensures the continuity of the target crop as opposed to destructive investigation. The technique involves data collection of discontinuous images and growth data for training. This growth data is collected using destructive investigation for training the model. Then, the trained image converter is used to estimate growth on continuous image datasets. A separate growth predictor uses the estimated growth data and environment data to predict growth.
Strengths:
1. The paper proposes a technique for non-destructive growth estimation of the lettuce crop and will be useful in indoor farming using controlled environment variables.
2. The use of a 2-step process enhances the explainability of the framework.
3. The authors explore multiple existing techniques to identify the most suitable model for the image converter and the growth predictor.
Weaknesses:
1. While Figure 2 provides a diagrammatic representation of how the predictor works, it is not clearly explained in the text. For example, why is it 144*7 (why 144)?
2. Figure 1 needs to be slightly modified to ensure that it is clear that training of the image converter is done on the discontinuous images.
3. The axes labels in figure 4 and Figure 7 need to be better described along with what each of the lines represent.
4. Figure 5: I am assuming that the number after DAT represents a dataset number, but that needs to be clearly mentioned.
5. This paper only uses lettuce for all its experiments and experimenting this model on different crops would make the paper more impactful.
Comments on the Quality of English Language
The paper needs to be reviewed by a bilingual or native English speaker to improve readability.
Author Response
Comments 1: While Figure 2 provides a diagrammatic representation of how the predictor works, it is not clearly explained in the text. For example, why is it 144*7 (why 144)?
Respones 1: The information has been described in lines 145-148. To predict the growth for the following day, the input data for the model consists of three estimated growth parameters from the previous model, environmental data, crop age, and 2-day intervals at 20-minute intervals. Thus, the input data size is 144 * 7. We add the explanation in caption of fig. 2.
Comments 2 : Figure 1 needs to be slightly modified to ensure that it is clear that training of the image converter is done on the discontinuous images.
Respones 2 : We added boxes with a dashed line for the training and verification steps to clarify input and output data (Fig. 1).
Comments 3 : The axes labels in figure 4 and Figure 7 need to be better described along with what each of the lines represent.
Respones 3 : We mistakenly omitted the relevant information, so we added an explanation about the lines in the caption for Figs. 4 and 7 (L202-203; L238-239).
Comments 4 : Figure 5: I am assuming that the number after DAT represents a dataset number, but that needs to be clearly mentioned.
Respones 4 : We added the full name of the DAT, days after transplanting (L229-230).
Comments 5 : This paper only uses lettuce for all its experiments and experimenting this model on different crops would make the paper more impactful.
Respones 5 : This study aimed to determine whether it is possible to effectively monitor and predict growth non-destructively using lettuce, the model plant, in a vertical farm module. In future research, we will consider the insights you provided to apply and evaluate this approach across various crops.
Reviewer 3 Report
Comments and Suggestions for Authors
The manuscript by Choi et al. is devoted to development of methods of lettuce growth monitoring and prediction. The work seems to be interesting; however, I have comments and questions.
1. Introduction: Is prediction of lettuce growth parameters on basis of mechanistic models of growth of these plants possible? What are classical and recent mechanistic models of lettuce growth? What are preferences of using machine learning in comparison with mechanistic models? It should be discussed in more detial.
2. Sections 2.2 and 2.3: Authors noted that “A total of 161 images and growth datasets were collected” (line 87) and “The number of augmented data was 1,224” (line 101). It is not clear: Howe were these quantities related? Were 1,224 images prepared from 161 images? Additionally, 161 images or, even, 1,224 images seem to be too small quantity for deep machine learning. Was this quantity enough?
3. Section 2.7: Why were 24 h photoperiod used for plant before 11 days and 16 h photoperiod used for plant after 11 days of the cultivation? There is experimental work (Jin et al., Gradually increasing light intensity during the growth period increases dry weight production compared to constant or gradually decreasing light intensity in lettuce. Sci. Hort. 2023, 311, 111807), which showed that increasing photoperiod with increasing age of lettuce plant improved productivity.
4. Figure 3, Supplementary table 1: Why were the determination coefficients (R2) negative? R2 described portion of experimental changes, which is described by model. How was this portion less than 0? It should be explained.
5. Figure 4: What was linear equation used for description of relations between experimental (?) and estimated (?) plant growth parameters? Estimated parameter = experimental parameter? Additionally, it was not clear: What did continuous and dotted lines in this figure mean?
6. Figure 7: What did continuous and dotted lines in this figure mean?
Author Response
Comments 1 : (Introduction) Is prediction of lettuce growth parameters on basis of mechanistic models of growth of these plants possible? What are classical and recent mechanistic models of lettuce growth? What are preferences of using machine learning in comparison with mechanistic models? It should be discussed in more detial.
Respones 1: This study focused on monitoring the growth changes of lettuce non-destructively for dynamic feedback control, while also emphasizing the prediction of lettuce growth through interactions with measured environmental factors. Therefore, rather than a mechanistic model, which generalizes the growth mechanism, machine learning is more suitable for dynamically estimating the state of lettuce non-destructively over time and providing feedback by predicting lettuce growth from these time points (lines 33-43).
Comments 2 : (Sections 2.2 and 2.3) Authors noted that “A total of 161 images and growth datasets were collected” (line 87) and “The number of augmented data was 1,224” (line 101). It is not clear: Howe were these quantities related? Were 1,224 images prepared from 161 images? Additionally, 161 images or, even, 1,224 images seem to be too small quantity for deep machine learning. Was this quantity enough?
Respones 2: Details on the relationship between the original image data and the augmented data have been added in lines 99 and 101. A total of 1,224 augmented images were generated, representing a 9-fold augmentation of the divided training dataset. To check if the model's training was subject to bias or overfitting due to limited data, the training data (first to third cultivation) and validation data (fourth to fifth cultivation) were collected from different cycles and locations (lines 92-94). The model successfully regressed the growth parameters from the validation data (lines 182-184).
Comments 3 : (Section 2.7) Why were 24 h photoperiod used for plant before 11 days and 16 h photoperiod used for plant after 11 days of the cultivation? There is experimental work (Jin et al., Gradually increasing light intensity during the growth period increases dry weight production compared to constant or gradually decreasing light intensity in lettuce. Sci. Hort. 2023, 311, 111807), which showed that increasing photoperiod with increasing age of lettuce plant improved productivity.
Respones 3 : The purpose of this study was not to identify environmental conditions that promote lettuce growth but rather to estimate time-series growth parameters from lettuce images within a vertical farm module and predict growth by utilizing concurrently measured environmental data. Therefore, this study was conducted in two stages (nursery and after trans-transplant) based on the light cycles commonly used in commercial vertical farm modules. In future research on developing a decision-making model for optimal lettuce growth through non-destructive methods, we will incorporate the insights you provided.
Comments 4 : (Figure 3, Supplementary table 1) Why were the determination coefficients (R2) negative? R2 described portion of experimental changes, which is described by model. How was this portion less than 0? It should be explained.
Respones 4 : A coefficient of determination (R-squared) less than zero indicates that the data may be arbitrary or that the model's performance is worse than predicting with the mean of the independent variables. In other words, an R-squared that minimizes the sum of residuals to zero is derived from the training data. However, since our study used test data collected from entirely different cultivation cycles for model validation, poorly trained models may show negative R-squared values when tested on such arbitrary test data.
Comments 5 : (Figure 4) What was linear equation used for description of relations between experimental (?) and estimated (?) plant growth parameters? Estimated parameter = experimental parameter? Additionally, it was not clear: What did continuous and dotted lines in this figure mean?
Respones 5-1 : The equations for the regression lines were added in the figures (Figs. 4 and 7). These two figures show accuracy from the training and verification of the image converter. That is, the accuracies in figure 7 were derived from the image converter having the same parameters (i.e., trained weights) with the fully trained image converter.
Respones 5-2 : The continuous and dotted lines represent the regression lines and the y = x lines, respectively. We added relevant information at the caption (L202-203; L238-239).
Comments 6 : (Figure 7) What did continuous and dotted lines in this figure mean?
Respones 6 :It represents y = x line for the comparison. We added relevant information at the caption (L238-239).
Round 2
Reviewer 3 Report
Comments and Suggestions for Authors
Authors completely considered my comments. I have not additional remarks or questions. The work seems to be interesting and perspective.